# Depression Pathophysiology: Astrocyte Mitochondrial Melatonergic Pathway as Crucial Hub

**DOI:** 10.3390/ijms24010350

**Published:** 2022-12-26

**Authors:** George Anderson

**Affiliations:** CRC Scotland & London, Eccleston Square, London SW1V 1PX, UK; anderson.george@rocketmail.com

**Keywords:** depression, mitochondria, melatonin, gut microbiome, astrocytes, aryl hydrocarbon receptor, amygdala, stress, racism, treatment

## Abstract

Major depressive disorder (MDD) is widely accepted as having a heterogenous pathophysiology involving a complex mixture of systemic and CNS processes. A developmental etiology coupled to genetic and epigenetic risk factors as well as lifestyle and social process influences add further to the complexity. Consequently, antidepressant treatment is generally regarded as open to improvement, undoubtedly as a consequence of inappropriately targeted pathophysiological processes. This article reviews the diverse array of pathophysiological processes linked to MDD, and integrates these within a perspective that emphasizes alterations in mitochondrial function, both centrally and systemically. It is proposed that the long-standing association of MDD with suppressed serotonin availability is reflective of the role of serotonin as a precursor for the mitochondrial melatonergic pathway. Astrocytes, and the astrocyte mitochondrial melatonergic pathway, are highlighted as crucial hubs in the integration of the wide array of biological underpinnings of MDD, including gut dysbiosis and permeability, as well as developmental and social stressors, which can act to suppress the capacity of mitochondria to upregulate the melatonergic pathway, with consequences for oxidant-induced changes in patterned microRNAs and subsequent patterned gene responses. This is placed within a development context, including how social processes, such as discrimination, can physiologically regulate a susceptibility to MDD. Future research directions and treatment implications are derived from this.

## 1. Introduction

Major depressive disorder (MDD) has been conceptualized in a wide array of diverse frames of reference over the millennia, including cognitive, psychoanalytic and biological, with a wide array of treatments thereby derived, usually with little more efficacy than placebo [1]. The clinical relevance of this is highlighted by the data showing the high lifetime prevalence of MDD (15–40%), especially as fewer than 50% of MDD patients show full remission from management with current antidepressants. This is further confounded by the data indicating that over 30% of MDD patients show no treatment response, often classed as suffering from treatment-resistant depression. This can have dire consequences given the percentages of people with severe MDD across all age groups that attempt or commit suicide [2].

### 1.1. MDD Pathophysiology

Classically, MDD pathophysiology has been strongly associated with the dysregulation of monoamines, particularly serotonin [3]. However, a diverse array of pathophysiological processes have been highlighted in recent research, including gut dysbiosis and gut permeability [4], oxidative and nitrosative stress [5], suboptimal mitochondrial/metabolic function [6], immune-inflammation [7], including mitochondrial dysregulation in immune cells [8], impaired neurogenesis [9], amygdala-frontal cortex alterations [10], early developmental events [11], the tryptophan/kynurenine ratio [12], the opioidergic system [13,14] and circadian rhythm dysregulation [15]. This article briefly reviews these broad bodies of data, highlighting how they may be intimately linked to alterations in astrocyte regulation of neuronal activity, as powerfully determined by the mitochondrial melatonergic pathway.

### 1.2. Stress, Resilience and the Amygdala

MDD is widely accepted as a heterogeneous condition, including in regard to its pathophysiology, with clinical investigations invariably finding that not all MDD patients show changes in the pathophysiological processes measured. MDD pathophysiology is further confounded by the presence of comorbidities, such as obesity, hypertension, chronic pain, anxiety, abusive/traumatic events, substance abuse, arthritis, and cardiovascular disorders, which adds another layer of complexity to the nature of depression’s biological underpinnings. Clearly, the utilization of signs and symptoms by current classification systems, rather than biologically defined criteria, can confound pathophysiological heterogeneity and contribute to the complexity of data interpretation in MDD research. MDD heterogeneity has undoubtedly contributed to a lack of appropriately targeted treatment [16].

MDD pathophysiological heterogeneity is also complicated by the effects of a variety of stressors, which have been traditionally modelled as being mediated via the hypothalamic-pituitary-adrenal (HPA) axis [17], as well as from the early direct effects of hypothalamic and amygdala corticotrophin releasing hormone (CRH). CRH acts on gut mast cells to induce tumor necrosis factor (TNF)α release, thereby increasing gut permeability [18] and inflammatory immune responses [19]. The classical association of stress with MDD susceptibility and resilience has primarily focused on alterations in neuronal function, neurotransmitter release and post-synaptic receptor activation [20], such as resilience-linked neuropeptide Y (NPY) [21]. However, more systemic processes are now known to regulate stress, including psychosocial stress-induced gut dysbiosis and gut permeability [22], which has consequences for brain patterned brain function [23].

Early developmental stressors allow the interaction of epigenetic processes and MDD susceptibility genes, including in the regulation of resilience [21]. A diverse array of stressors can act prenatally [24], early postnatally [25] and in childhood [26,27], driving stress-sensitization that can form a more trait-like MDD susceptibility, namely Neuroticism [28] and associated attenuation in stress resilience [29]. Such developmental stressors do not seem neuropsychiatric disorder specific [30,31,32,33], although all associate with MDD susceptibility and possibly accelerated aging [34]. Early developmental stressors partly mediate their impact via the developing gut, and the consequences that this has for amygdala development [35,36,37], via the early developmental role of the amygdala on cortex development [38].

Most investigations of amygdala function focus entirely on neuronal activity. However, recent data shows a significant role for alterations in astrocyte function and associated neuronal regulation in the amygdala, as in other brain regions. Basolateral amygdala astrocytes significantly contribute to aversive/fear learning via driving variations in the activity of amygdala-medial prefrontal cortex (mPFC) interactions [39], as well as depression-associated behaviors in preclinical models [40]. Chronic unpredictable stress in rodents induces behavioral anhedonia via medial amygdala astrocyte effluxes [41], as well as via κ-opioid receptor activation, the activation of which induces dysphoria [42]. Astrocytes are also powerful determinants of motivated behaviors via their regulation of the nucleus accumbens (N.Acc) and the N.Acc interactions with PFC, basolateral amygdala, and ventral hippocampus inputs [43]. Overall, the predominant neuron-centric conceptualizations of amygdala function in stress and mood regulation is shifting to one focusing on astrocyte-neuronal interactions.

### 1.3. Mitochondria

Metabolic dysregulation and suboptimal mitochondrial function across a wide array of CNS and systemic cells is common in MDD [44,45,46]. The intravenous injection of mitochondria prevents LPS-induced depressive behaviors in rodents, as well as increasing neurogenesis, brain-derived neurotrophic factor (BDNF), ATP and sucrose preference, whilst decreasing astrocyte and microglia ROS and pro-inflammatory cytokine induction [47]. Mitochondria are the major source of ROS in cells, with ROS increased under challenge, including from suppressed endogenous antioxidants. Optimized mitochondrial function is powerfully determined by the pyruvate dehydrogenase complex (PDC) and the PDC conversion of pyruvate to acetyl-CoA, with PDC also a significant regulator of ROS production [48,49]. Mitochondrial ROS is an important form of signaling and has numerous downstream consequences, including ROS-induced microRNAs [50], and therefore patterned gene expressions. Cutting edge interest in the role of miRNAs in MDD [51], and many other diverse conditions may therefore be intimately linked to mitochondrial function.

As astrocytes are powerful determinants of neuronal function, neurotransmitter levels and survival, there is a growing interest in astrocyte mitochondria, including as to how mitophagy and mitochondrial biogenesis form core aspects of astrocytes mitigation of oxidative stress [52]. Although classically regarded as glycolytic cells, recent data shows the suppression of astrocyte mitochondrial oxidative phosphorylation (OXPHOS) to dramatically alter astrocyte shape, function and neuronal regulation [53]. Pyruvate is an important astrocyte energy substate, being converted to acetyl-CoA, which optimizes ATP production from the tricarboxylic acid (TCA) cycle and OXPHOS, whilst also providing a necessary co-substrate for the conversion of serotonin to N-acetylserotonin (NAS) in the initiation of the melatonergic pathway [54]. Importantly, astrocyte mitochondria also participate in neuronal redox homeostasis as well as in the recycling of neurotransmitters [55]. Such data highlights how the alterations in mitochondrial function that are evident centrally and systemically in MDD, will have important effects on metabolism in astrocytes and thereby on neuronal activity regulation.

### 1.4. Neurotransmitters and Antidepressants

Current conceptualizations of MDD pathophysiology have been disappointing in providing quick and effective anti-depressant treatments [56], with most clinical focus being on released monoamines [3], which are all primarily defined by their capacity to regulate released neurotransmitter levels [57,58,59,60]. This includes the quick-acting antidepressant effects of (es)ketamine, which is typically defined as a neuronal N-methyl-d-aspartate (NMDA) receptor antagonist [57,61], although also acting via the opioidergic system [13]. Other CNS neuronal processes, such as synaptogenesis/synaptoplasticity and neurogenesis [62] have also been intimately linked to MDD pathophysiology and treatment response, although glial regulation can underpin such processes [63].

Recent data indicates that traditional and novel antidepressants mediate their effects via astrocytes, including via the upregulation of astrocyte mitochondria ATP, with subsequent consequences for neuronal activity and transmission [64], as well as via the regulation of gap junctions forming the astrocytic network [65]. Such data highlights how the conceptualization of MDD as a form of glia-neuronal interactions, powerfully determined by variations in astrocyte function can initiate the development of new treatments.

### 1.5. Pro-Inflammatory Cytokines, Tryptophan and Kynurenine

Systemic processes, including the immune system [1] and gut microbiome/permeability [66] have long been linked to MDD pathophysiology. Alterations in the levels, phenotypes and function of an array of diverse immune cells have been associated with MDD pathophysiology, including monocytes/macrophages [67], dendritic cells [68], t-helper (Th)17 cells [69], enteric glial cells [70], CD8+ t cells [71] and natural killer (NK) cells [72], usually in association with heightened systemic levels of pro-inflammatory cytokines and oxidative-nitrosative stress [73,74].

Pro-inflammatory cytokines and ROS increase indoleamine 2,3-dioxygenase (IDO) and tryptophan 2,3-dioxygenase (TDO), which convert tryptophan to kynurenine, thereby depleting tryptophan availability for serotonin and melatonin synthesis, whilst increasing neuro- and glia-regulatory tryptophan products, including the opposing effects of kynurenic acid (KYNA) and quinolinic acid (QUIN) at the NMDAr [75,76]. Kynurenine metabolism may be differentially regulated at different CNS sites, with an increase in the KYNA/QUIN evident in the anterior cingulate in MDD patients, coupled to heightened levels of EAAT2, indicating significant changes in astrocyte regulation of glutamatergic transmission in particular CNS subregions [77], whilst heightened microglia-derived QUIN levels are evident in the subgenual anterior cingulate cortex of MDD patients committing suicide [78]. Altered kynurenine pathway activity is clearly evident in different brain regions, where it is powerfully determined by glia, with significant consequences for patterned interarea communication. Kynurenine pathway activation drives down levels of serotonin, NAS and melatonin, which are all decreased in MDD patient brains [79], as in other medical conditions [80]. Glia-driven changes in cytokines-ROS/IDO-TDO/kynurenine-tryptophan-melatonin regulation are intimately linked to the alterations in patterned neuronal activity in the CNS in MDD.

### 1.6. Aryl Hydrocarbon Receptor

Although kynurenine and kynurenic acid also activate the aryl hydrocarbon receptor (AhR), there is a paucity of studies on the AhR in MDD. The AhR is activated by endogenous and exogenous ligands, thereafter having significant impacts on the functioning of immune cells, including brain astrocytes [81] and microglia [82]. The AhR is also activated by gut microbiome-derived products, including indole-3-acetate [83], whilst AhR activation can induce stress-associated CRH [84,85]. Kynurenine activation of the AhR can drive mitochondrial oxidative stress and neuronal apoptosis [86]. The AhR is also expressed on the mitochondria membrane, where it can increase Ca2+ influx via the voltage-dependent anion channel (VDAC)1 [87]. Many of the immune-associated changes in MDD may be mediated by heightened AhR activation, as in other medical conditions [88,89,90]. The AhR has complex effects, including differential consequences arising from different AhR ligands.

The AhR is in negative reciprocal interactions with melatonin. AhR-induced cytochrome P450 (CYP)1B1 not only metabolizes melatonin but also ‘backward’ converts melatonin to NAS. Some studies show the AhR to suppress 14-3-3, suggesting that it may have wider impacts on the melatonergic pathway, and thereby attenuating the deactivation of astrocytes and microglia [1]. The differential regulation of the kynurenine pathway by astrocytes and microglia in different brain subregions will be intimately linked to the differential activation of the AhR in these subregions, with consequences for neuronal metabolism and function.

### 1.7. Gut Microbiome, LPS and Butyrate

Gut dysbiosis and gut permeability drive mucosal and systemic immune responses, to form the ‘gut-immune-brain’ axis in MDD pathophysiology [91]. Gut permeability increases circulating lipopolysaccharide (LPS), which has direct effects on CNS cells and indirect effects via systemic immune cells. LPS activation of Toll-like receptor (TLR)4, like other gut microbiome-derived products, are commonly modelled as having direct effects on neuronal function and survival [92]. However, the effects of LPS, and the gut microbiome-derived epigenetic regulator, butyrate, have direct effects in brain glia, as well as indirect effects via the systemic immune system. As the gut microbiome/permeability is now a cutting-edge area of research across most medical conditions, this is covered in more detail below.

### 1.8. Ceramide

Circulating ceramide is significantly associated with MDD in clinical and preclinical models [93], mediated via ceramide acting on brain endothelial cells to inhibit phospholipase D (PLD), thereby suppressing phosphatidic acid and increasing hippocampal phosphotyrosine phosphatase (PTP1B). As PTP1B is an important determinant of tyrosine phosphorylation of an array of cellular proteins, including the BDNF receptor, TrkB, circulating ceramide is proposed to inhibit dentate gyrus neurogenesis and drive wider MDD pathophysiology [94]. Circulating ceramide can have a number of sources, including high-fat diet exposed intestinal epithelial cells [95], and platelet activation [96], as well as from other brain cells, including activated astrocytes and microglia [97]. Astrocyte activation and ceramide efflux can also damage white matter, by enhancing demyelination and suppressing remyelination, whilst concurrently increasing BBB permeability [98]. This is one of the mechanisms through which glia-linked MDD pathophysiology can contribute to exacerbations in neurodegenerative conditions.

### 1.9. Social Processes and Discrimination

A relatively neglected aspect of MDD pathoetiology and pathophysiology are the consequences of societal social stratification that is differentially evident across different cultures, including social class as well as racial and sexual discrimination, and their complex interface with poverty and increased exposure to domestic and occupational environmental toxins. A host of current classifications for an array of diverse medical conditions show susceptibility and/or outcomes to be modulated by self-identified race and experiences of racial discrimination, including MDD susceptibility [99,100,101]. The association of pathophysiology with social stratification is a complex and controversial area. Social stressors may parallel more investigated stressors by increasing gut dysbiosis/permeability and IDO/kynurenine/AhR activation, thereby decreasing natural killer cell and CD8+ t-cell cytotoxicity, whilst enhancing macrophage pro-inflammatory responses leading to a dysregulated immune response, with systemic and central consequences [90,102]. Some data is supportive of this [103]. This is clearly an area needing extensive investigation and is likely to involve general stress and resilience processes involving astrocyte regulation of amygdala-PFC interactions and alterations in the opioidergic system as indicated above in Section 1.2.

### 1.10. Circadian Dysruption

Circadian rhythm disruption associates with diverse social and medical conditions [104,105,106], including MDD [107], typically involving dysregulated cortisol levels and suppressed pineal melatonin, inducing the loss of the nocturnal dip in blood pressure [108], in the course of hypertension development [109]. As pineal melatonin dampens residual inflammatory activity in immune and glia cells [110], at least partly via the optimization of mitochondrial function, this is another aspect of MDD pathophysiology that can be mediated via astrocyte regulation of neuronal activity.

### 1.11. Summary

Much of the work on MDD pathophysiology has investigated alterations in neuronal function and neurotransmitter release. However, more recent conceptualizations of brain function have emphasized the roles of glial cells, especially astrocytes in the regulation of neuronal function, neurotransmitter release and neuronal survival [111]. In this context, neuronal activity may be conceptualized as a form of immune-to-immune communication, with astrocytes and microglia being the most common brain immune cells [38,112]. The current article looks to better integrate the above diverse and disparate cutting-edge areas of MDD research by highlighting the roles of the above factors on the interface of astrocytes with neurons, and the powerful role of the astrocyte mitochondrial melatonergic pathway in determining patterned neuronal activity across the brain.

## 2. Integrating MDD Pathophysiology

The above diverse bodies of data on MDD pathophysiological underpinnings are challenging to integrate, which is undoubtedly complicated by the pathophysiological heterogeneity of MDD, as currently defined in the absence of any biomarkers. The current article looks to integrate the above by highlighting an important role for astrocytes as a crucial hub on which many processes act to drive changes in mood and affective regulation. As such, understanding the role of astrocyte-neuronal interactions in MDD requires the integration of wider body processes that act upon this crucial hub.

### 2.1. Gut Microbiome, Butyrate and Mitochondria, ROS and microRNAs

As indicated above, MDD pathophysiology involves a wide array of diverse factors and processes, including two gut processes, namely: (1) gut permeability-induced circulating LPS and activation of the TLR4/ Nuclear factor kappa-light-chain-enhancer of activated B cells (NF-kB) /yin yang (YY)1 pathway; and (2) decreases in the short-chain fatty acid, butyrate and therefore attenuation of butyrate’s epigenetic regulation via histone deacetylase (HDAC) inhibition as well as the loss of butyrate’s mitochondria-optimizing effects via upregulation of the sirtuin-3 deacetylation and disinhibition of PDC. The conversion of pyruvate to acetyl-CoA by PDC not only increases ATP production from the tricarboxylic acid (TCA) cycle and OXPHOS, but also makes acetyl-CoA available as a necessary co-substrate for the initiation of the melatonergic pathway. The latter may be of some importance, given that the antioxidant and glutathione (GSH)-inducing effects of intracrine and autocrine melatonin suppress the PDC induction of ROS [113]. Increased LPS and suppressed butyrate levels are two ways in which the gut microbiome/permeability act to regulate cellular and mitochondrial function across systemic and central cells, with important consequences for reactive cells, including glia and immune cells. Within the CNS such gut microbiome-driven effects have been mostly characterized, to date, as occurring in microglia, rather than astrocytes [23]. As a HDAC inhibitor, butyrate also increases the astrocyte melatonin MT1 receptor [114], indicating that variations in gut microbiome-derived butyrate will modulate the levels and effects of autocrine melatonin in astrocytes and other cells across the body. See Figure 1.

The butyrate optimization of mitochondrial function requires the melatonergic pathway [115], via butyrate/sirtuin-3/PDC providing the acetyl-CoA that is a necessary co-substrate for the conversion of serotonin to N-acetylserotonin (NAS) and thereafter melatonin. The loss of butyrate-induced melatonin or the suppression of the melatonergic pathway via a decrease in tryptophan availability, tryptophan uptake, 14-3-3ε stabilization of tryptophan hydroxylase (TPH)2, 14-3-3ζ stabilization of arylalkylamine N-acetyltransferase (AANAT) and/or acetyl-CoA will decrease mitochondrial antioxidant status, including via the loss of melatonin/GSH induction [113]. The suppression of endogenous antioxidants markedly increases oxidants produced by the PDC [116]. Given that mitochondrial ROS is a major driver of oxidant regulation of patterned microRNAs (miRNAs) via miRNA biogenesis, transcription, and epigenetic modifications, a significant change in the patterning of miRNAs and associated changes in gene patterning will occur [50,117]. Gut microbiome/permeability factors therefore have significant impacts on patterned gene expressions, and their intra- and inter-cellular consequences, via the mitochondria melatonin/ROS/miRNAs pathway, forming the underpinnings of why variations in the gut microbiome/permeability significantly impacts on seemingly all medical conditions, including MDD.

Key sites where such changes will have significant impacts in the development of mood dysregulation, include in the mitochondria of enteric glial cells, intestinal epithelial cells, CNS astrocytes and microglia, as well as in systemic immune cells and circulating platelets. This gut microbiome/permeability influence does not emerge in adulthood, but seems evident from early development, as indicated by the data showing the gut microbiome/permeability to regulate amygdala development, and therefore contributing to developmental variations in emotional regulation of brain area interactions underpinning mood disorders. It is by such processes that the gut modulates astrocyte-neuronal interactions in the pathophysiology of MDD. Recent data highlighting the role of the opioidergic system in MDD may be linked to this [13].

### 2.2. Gut Microbiome, Amygdala and Opioidergic System

Alterations in the opioidergic system, especially the μ-/κ-opioid receptor ratio and their ligands, β-endorphin and dynorphin, respectively, are a cutting-edge area of research in MDD pathophysiology [13,118]. The gut microbiome significantly modulates the opioidergic system, including via butyrate’s upregulation of the μ-opioid receptor [119]. The μ-opioid receptor is intimately linked to social bonding, including from oxytocin’s partial μ-opioid receptor agonist effects [120]. The butyrate induction of mitochondrial melatonin upregulates the endogenous μ-opioid receptor ligand, β-endorphin [121], indicating that some of the beneficial effects of butyrate may be dependent upon its capacity to upregulate cellular melatonin. In contrast, κ-opioid receptor activation in the amygdala induces dysphoria [122]. These regulatory effects of butyrate on the opioidergic system are intimately associated with its capacity to optimize mitochondrial function and upregulate the glia mitochondrial melatonergic pathway, including in the amygdala and associated paracapsular cells of the intercalated masses [38,123], thereby regulating amygdala-PFC interactions.

The amygdala inputs to the ventral tegmental area (VTA)-N.Acc junction are an important determinant of affect-driven motivated behaviors, with the opioidergic system also a significant direct regulator of VTA-N.Acc associated motivated behaviors [124]. The HDACi effects of butyrate in the N.Acc significantly upregulates motivated behaviors, driven by μ-opioid receptor activation [125]. The κ-opioid receptor in these brain reward regions correlates negatively with social status [126]. Such data highlights how the gut can act on opioidergic processes in different brain regions in the modulation of MDD pathophysiology. It is important to note that such gut regulation of the opioidergic system in the amygdala and VTA-N.Acc will be determined by effects on astrocyte-neuronal interactions, involving alterations in the mitochondrial melatonergic pathway in these different brain regions, and will be important aspects of the early developmental processes regulating such inter-area communication across brain regions [127,128]. Importantly, the amygdala can over-ride cortex and hippocampal inputs into the VTA-N.Acc junction, as shown in rodents, highlighting the relative strength of affect over higher order cognition in the regulation of motivated behavioral outputs [129,130].

The amygdala and VTA-N.Acc junction are intimately linked to MDD pathophysiology. However, it is the regulation of astrocytes and the astrocyte mitochondrial melatonergic pathway that is a crucial determinant of astrocyte-neuronal interactions at these important hubs in MDD pathophysiology. Gut-derived factors such as LPS and butyrate act on MDD pathophysiology at these brain sites via their impact on astrocyte mitochondrial function and thereby on astrocyte-neuronal interactions.

### 2.3. Gut Microbiome and Ceramide

The significance of circulating ceramide in MDD will be important to clarify in future research [93,94]. Ceramide can be derived from many cellular sources [96,97,131], including astrocytes [98], contributing to the association of MDD with neurodegenerative conditions via suppressed mitochondrial function [98,132]. ROS and proinflammatory cytokines induce ceramide-producing enzymes, leading to ceramide’s inhibition of the mitochondrial respiratory chain and suboptimal mitochondrial function [132]. This allows ready links to alterations in the gut microbiome/permeability and the mitochondrial melatonergic pathway, given their regulation of ROS and pro-inflammatory cytokines.

Gut dysbiosis/permeability and decreased butyrate in MDD [133] raises levels of gut-derived, circulating trimethylamine N-oxide (TMAO). Heightened TMAO is evident in MDD [134] and activates platelets, which further contribute to ceramide production and release, and to the association of MDD with other medical conditions [135]. Importantly, gut microbiome-derived butyrate suppresses ceramide levels via butyrate converting ceramide to glucosylceramide, which then acts as a precursor for an array of gangliosides [136]. Butyrate, like melatonin, also suppresses platelet activation [98] and microglia activation [137], indicating the importance of the gut microbiome in regulating the ceramide levels via a number of different routes across different cell types. It will be important to determine as to whether the effects of butyrate in suppressing ceramide require the induction of the mitochondrial melatonergic pathway, including in platelets, endothelial cells and systemic immune cells as well as glia, and the consequences that this has on astrocyte-neuronal interactions. Such data will contribute to an understanding of the heterogeneity of MDD pathophysiology, and how systemic cells and their interactions act to modulate astrocyte-neuronal interactions.

### 2.4. Gut Microbiome, Amygdala and Personality

Preclinical data shows that exposure to the gut microbiome-derived short-chain fatty acid, propionate, in adolescence produces significant changes in the mitochondria of amygdala astrocytes, microglia and neurons with immediate changes in social behavior [138], with relevance to high addiction comorbidity [139] and the development of personality traits linked to MDD susceptibility [140,141]. Such data highlights the diversity and complexity of data pertaining to MDD and the challenge, and opportunity, that such data provides in finding a parsimonious physiological model of human affect and trait development.

### 2.5. Neurogenesis and the Astrocyte Mitochondrial Melatonergic Pathway

Suppressed neurogenesis is a classical aspect of MDD pathophysiology. Neurogenesis involves the interactions of astrocytes and neuronal progenitor cells in the dentate gyrus, including via astrocyte BDNF activation of TrkB. Notably, TrkB is also activated by NAS [142], indicating that the backward conversion of melatonin to NAS, including by activation of the purinergic receptor P2Y1, metabotropic glutamate receptor, mGluR5, and the AhR [143], will increase NAS production and release [63], with NAS activation of TrkB increasing neurogenesis [144]. NAS also induces hippocampal BDNF [145]. As such, the astrocyte mitochondrial melatonergic pathway provides response plasticity that can be influenced by local and systemic processes. As to how circulating ceramide suppression of neurogenesis in MDD [96] interacts with the astrocyte mitochondrial melatonergic pathway requires investigation.

### 2.6. Integrating Wider Bodies of Biomedical Data

MDD pathophysiology has been linked to diverse bodies of data, including discrimination stress (see Figure 2), at least in part by its association to a wide range of environmental and social factors that ultimately act on physiological interactions across the body over the course of development. Such a ‘holistic’ perspective also has relevance for the early developmental pathoetiology of a host of adult-onset medical conditions, including Parkinson’s disease, amyotrophic lateral sclerosis (ALS), multiple sclerosis and rheumatoid arthritis, as well as to how MDD interacts with, and may trigger exacerbations in, such conditions [146,147,148,149]. Engaging the complexity of MDD therefore has consequences for the conceptualization and treatment of diverse medical conditions that are currently poorly managed.

The wide range of data reviewed above and their impact on glia-neuronal interactions provide an integrated framework for understanding MDD pathophysiology within a developmental frame of reference, upon which many genetic and epigenetic factors may act. This framework is derived from an assumption that the evolutionary modified bacteria in the form of mitochondria that drive energy provision in almost all body cells are important determinants of cellular function, including patterned miRNA and gene expressions. This allows intercellular interactions to be conceived, at least partly, as a form on mitochondria-to-mitochondria communication, with the diverse and dynamic changes occurring in cellular fluxes forming the means of such communication. The intimate association of the melatonergic pathway with mitochondria, seemingly from the very first multicellular organism [150], and its maintained presence across the three kingdoms of life (animals, plants and fungi) over the course of 2–2.5 billion years of evolution strongly indicates that the mitochondrial melatonergic pathway is a core physiological process. The suboptimal mitochondrial function evident across systemic and central cells in MDD and the long-standing association of decreased serotonin in MDD may be framed within this context, given the necessity of serotonin as a precursor for the initiation of the melatonergic pathway. Cellular interactions may thereby be conceived, at least partly, as a form of modified bacteria communication with homeostasis being dependent upon all interacting cells being able to optimize their mitochondrial melatonergic pathway activity. Ageing, and factors that dysregulate the capacity of cells to maintain more optimal function of their melatonergic pathway, lead to alterations in the homeostatic status quo, which is the essence of neuroprogression across psychiatric conditions. Higher order body and brain systems develop within this basic framework and are shaped by the specific cell types in which mitochondria reside.

Within this context, the bacteria of the gut, lung, skin, and placental microbiomes are in interaction with their evolutionary modified distant cousins in the form of mitochondria. This has parallels to recent thinking on the tumor microenvironment, where cancer metabolism acts to regulate mitochondrial function, and the melatonergic pathways, in the other tumor microenvironment cells [151], allowing cancer cells to dominate the nature of local processes and therefore form a new homeostasis. As to whether such mitochondria/metabolic dominance occurs in other collections of body cells will be interesting to determine, including as to whether this drives a new homeostatic regulation, such as in subregions of the anterior cingulate in MDD. This provides a different conceptualization of aging, and how aging positively associates with almost all medical conditions. Within the tumor microenvironment such ‘metabolic dominance’ is achieved, at least partly, by IDO induction and the release of kynurenine, which activates the AhR in natural killer cells and CD8+ t cells, leading to the induction of a metabolic state of ‘exhaustion’, whereby these cells lose their capacity to kill cancer cells and virus-infected cells [152]. Another readily achievable means by which ‘metabolic dominance’ may be achieved is via the induction or exosomal/vesicular transfer of miRNAs that suppress the melatonergic pathway, such as the suppression of 14-3-3 by miR-7, miR-375, miR-451 and miR-709 [153,154,155], with consequences for the cells in which these miRNAs are induced, including melatonergic pathway suppression and alterations in the regulation of ROS, ROS-driven miRNAs and therefore patterned gene transcriptions. miR-709 also suppresses mitochondrial transcriptional factor A (TFAM) [156], although with some distinct effects in different cell types, indicating that wider aspects of mitochondrial function will be coordinated with alterations in melatonergic pathway regulation [157]. Importantly, such miRNA-suppression of 14-3-3 isoforms and the melatonergic pathway will have detrimental effects on the capacity of butyrate to optimize mitochondrial function, indicating that the beneficial effects of butyrate are intimately intertwined with the capacity of a given cell to regulate the melatonergic pathway. In the case of a suppressed mitochondrial melatonergic pathway, butyrate may not then be able to restore the homeostatic status quo. Aging-associated increases in miR-709 in the murine liver suppresses hepatic function [158], suggesting that the suppression of the mitochondrial melatonergic pathway may be a significant aspect of organ dysfunction over aging, including in the liver but also in other organs.

This indicates a resetting of local interactions that is coupled to a suppressed capacity of gut microbiome-derived butyrate to return these interactions to their previous homeostatic state. This would also suggest a suppressed capacity of circadian pineal melatonin to reset mitochondrial function at night, which would be further contributed to by the dramatic decrease in pineal melatonin over the course of aging. It is generally assumed that over the course of recurrent depression episodes, there is a change in the homeostatic regulation, whereby even in the successful resolution of a depressive episode there is no return to the previous homeostatic state. This is the essence of the concept of neuroprogression in many psychiatric conditions, including MDD [159], and requires investigation as to pathophysiological alterations occurring in ‘treatment-resistant’ depression.

Although alterations in the mitochondrial function of many cells, including immune, glial, platelets and intestinal epithelial cells, as well as different neurons may occur in MDD, it is proposed here that astrocytes may be of particular relevance and act as an important hub onto which many of the above developmental, genetic, environmental and social processes may ultimately act.

## 3. Astrocytes as a Crucial Hub

For decades astrocytes were regarded primarily as providers of antioxidants and energy substrates for neurons. This has gradually changed over recent decades with astrocytes now recognized as powerful determinants of neuronal function and survival as well as of neurotransmitter release [160]. An astrocyte’s processes make contact with many synapses across different neurons, whilst also being part of an astrocyte network that communicates via a number of factors, including Ca2+ and ATP transfer across connexin (Cx)-43 gap junctions. Cx-43 also exist as hemichannels, being a route for astrocyte fluxes. Astrocyte end-feet are an integral part of the blood–brain barrier (BBB) in association primarily with vascular endothelial cells and pericytes. Astrocytes therefore provide a powerful interface for neuronal activity, coordinated blood flow regulation and systemic factors crossing the BBB. The astrocyte-neuronal interface would seem to be an important hub in all neuropsychiatric disorders. However, the specificity of astrocyte-neuronal interactions in MDD would seem to be determined by particular interactions across brain regions, such as amygdala and cortex, over the course of development, and the influence of the gut microbiome on such interactions.

Astrocytes have a number of immune-like qualities, being reactive cells with some antigen presenting capacity [161]. Maintained activation of a reactive state in astrocytes leads to the retraction of their processes from synapses and their isolation from the astrocytic network, with a maintained reactivated astrocytic state dysregulating wider coordinated patterned neuronal activity. Astrocytes release a wide array of factors that can be protective or damaging to neurons [97]. Astrocytes may not only regulate but also drive neuronal activity, including via the release of glutamate, which may be targeted to presynaptic and postsynaptic glutamatergic receptors. Astrocytes can release glutamate via a number of mechanisms, including Cx-43 hemichannels, vesicles and via the cystine-glutamate antiporter (System Xc), which effluxes glutamate in exchange for cystine uptake in the course of glutathione (GSH) synthesis, suggesting that increased antioxidant synthesis demand in astrocytes is intimately linked to glutamate efflux and the regulation of neuronal function.

GSH is an important inhibitor of neutral sphingomyelinase (nSMase)2-induced ceramide and pro-inflammatory cytokine release from astrocytes, with powerful consequences for neuronal function and associated cognition, as well as neuronal survival, as shown in preclinical models [162]. The relevance of nSMase2-induced ceramide is highlighted by the clinical and preclinical data showing stress-induced MDD to be significantly determined by circulating levels of nSMase2-induced ceramide [96]. Although these authors indicate brain endothelial cells as the major cell to be impacted by blood-derived ceramide in MDD, such data suggests the potential importance of variations in astrocyte oxidant status and fluxes on neuronal activity and interarea neuronal patterning, as well as on endothelial function, including via the capacity of GSH to suppress nSMase2-induced ceramide.

Astrocytes can be activated by many factors, the most extensively investigated being LPS via TLR4 activation, leading to NF-kB and YY1 transcription factors induction, as in many other cell types [163,164,165]. NF-kB is the transcription factor most strongly associated with astrocyte reactivation. NF-kB induction typically leads to a transient astrocyte activation state, including the induction of beta-site amyloid precursor protein (APP) cleaving enzyme-1 (BACE1) and subsequent production of amyloid-β [166]. Although classically associated with Alzheimer’s disease, BACE1 and amyloid-β are increased in a wide range of inflammatory conditions, including glioblastoma, breast cancers, amyotrophic lateral sclerosis and Parkinson’s disease with Lewy Bodies [167,168], as well as possibly in later-life MDD [169]. Given that amyloid-β is an endogenous antimicrobial, it’s release following NF-kB and YY1 induction seems predominantly to be an attempt to dampen microbial signaling, including as arising from TLR4 receptor activation by LPS, and perhaps endogenous ligands such as HMGB1 and hsp70. As such, processes such as gut permeability-derived LPS/TLR4/NF-kB signaling in MDD are intimately linked to changes occurring in astrocytes during neurodegenerative conditions. Alterations in astrocyte reactive states in MDD are therefore linked to a diverse array of medical conditions, including the risk and exacerbation of neurodegenerative conditions, although this is highly likely to be confounded by MDD pathophysiological heterogeneity.

In 2007, Liu and colleagues showed astrocytes to produce melatonin in a non-circadian manner [170]. In microglia and macrophages, the NF-kB induction of a reactive M1-like phenotype is concurrently linked to the induction of the melatonergic pathway and subsequent melatonin releases, leading to autocrine effects that induce an M2-like pro-phagocytic phenotype [171,172]. This allows the initial activation of these immune cells to be quickly followed by the production, release and autocrine effects of melatonin, thereby time-limiting the damaging effects that would arise from prolonged activation of these reactive cells. As to whether the NF-kB and YY1 induction of a reactive state in astrocytes is likewise sequentially associated with the upregulation of the melatonergic pathway, as indicated by NF-kB [165] and YY1 [173] in other cell types, requires investigation, including as to the relevance of this in different brain regions. Exogenous melatonin clearly dampens inflammatory activity in astrocytes [173], as in other reactive cells, with effects that seem, at least in part, via the capacity of melatonin to upregulate the mitochondrial melatonergic pathway, thereby better optimizing mitochondrial function and decreasing mitochondrial oxidants [174], with associated impacts on ROS-dependent miRNAs and gene patterning. It should also be noted that astrocyte YY1 has a number of important physiological effects, including over the course of normal development in mice, with some differential effects on gene induction in different brain regions [175]. It requires investigation as to whether these regulatory effects of YY1 are dependent upon its capacity to induce the melatonergic pathway. This is parsimonious with the detrimental effects of YY1 in a number of diverse medical conditions, including cancers [176], where the melatonergic pathway is dysregulated [145].

### Mitochondrial Melatonergic Pathway

Melatonin seems to be produced in all mitochondria-containing cells across the three kingdoms of life on earth [150], with the majority of melatonin being produced within mitochondria [177,178]. This is likely to be of significance in MDD pathophysiology. See Figure 1.

In the absence of the direct availability of serotonin from its release by serotonergic neurons and uptake into glia, the activation of the melatonergic pathway requires tryptophan uptake via the large amino acid transporter (LAT)-1 (SLC7A5), as well as SLC7A7 and SLC7A8. In astrocytes, as in many other cell types, the conversion of tryptophan to serotonin requires the induction of tryptophan hydroxylase 2 (TPH2), with TPH2 requiring the presence of the 14-3-3ε (YWHAE) isoform in order to be stabilized in an enzymatically active form [179]. The serotonin produced can then be converted by 14-3-3ζ (YWHAZ)-stabilized AANAT to NAS, which is then converted by acetylserotonin methyltransferase (ASMT) to melatonin [155]. For 14-3-3ζ-stabilized AANAT to initiate this pathway requires the presence of acetyl-CoA as a necessary co-substrate. The exclusivity of these 14-3-3 isoforms in the regulation of the melatonergic pathway in glia requires investigation. An array of diverse factors and processes may interact with these components of the melatonergic pathway, leading to a prolonged astrocyte activation state, dysregulated synaptic activity and neuronal oxidant challenge, as well as impacting on BBB homeostasis and the mitochondrial function of astrocytes, leading to changes in ROS-dependent miRNAs and gene patterning, with consequences for alterations in neuronal function, neurotransmitter release and thereby changes in inter-area communication.

YYI is also an important negative regulator of the excitatory amino acid transporter (EAAT)2, and therefore of glutamate uptake at the synaptic cleft. It requires investigation as to how a suppressed capacity of YY1 to upregulate the mitochondrial melatonergic pathway impacts on such neurotransmitter regulation. Alterations in glutamate regulation and the excitatory/inhibitory balance are an area of extensive research in MDD [180] and many other classical CNS conditions, such as autism, dementia, multiple sclerosis, ALS and schizophrenia [181,182,183,184]. YY1 is highly regulated by HDAC effects at the promotor of many YY1-induced genes, indicating that the loss of gut microbiome-derived butyrate’s HDACi capacity will have significant consequence for astrocyte YY1 regulation of the excitatory/inhibitory balance, which may be further dysregulated by suppression of the melatonergic pathway. It is still unknown as to whether the ten-fold decrease in pineal melatonin release at night between the ages of 18 years and 80 years are replicated in other cell types over the course of aging. Given the consequences that this can have on mitochondrial function and intercellular communication, as highlighted above, this should be a priority area of research in astrocytes, as well as other cell types.

Numerous studies have linked MDD, especially treatment-resistant MDD, to an array of autoimmune disorders [185], with MDD also showing evidence of autoimmune-linked processes [186]. Investigation of brain tissue indicates that genetic susceptibility to MDD is associated with a number of genes located in the major histocompatibility complex (MHC) locus of chromosome 6, providing a ready link of MDD to autoimmunity [187,188]. Neuroticism, a trait-like aspect of MDD susceptibility, is similarly associated with MHC-I [189]. MHC-I is increased by oxidative stress in brain cells, including astrocytes, neurons and oligodendrocytes, thereby increasing autoimmune activation, which can include the chemoattraction of CD8+ t cells [190]. The suppression of the mitochondrial membrane-located PTEN-induced kinase 1 (PINK1), not only attenuates mitophagy of dysfunctional mitochondria, but also increases oxidative stress and MHC-1, indicating that alterations in mitochondrial function may be an important mediator of the association of MDD and autoimmunity. Suppressed PINK1 levels are evident in MDD and MDD-associated pathophysiological changes, being an aspect of suppressed neurogenesis [191]. Importantly, exogenous melatonin promotes PINK1 accumulation on the mitochondrial membrane, as shown in neurons [192], indicating that melatonin not only upregulates the beneficial effects of PINK1 regarding mitophagy and protection against oxidative stress, but would also prevent the consequences of suppressed PINK1, including MHC-1 upregulation and associated induction of autoimmune-linked processes [193]. Astrocyte MHC-I is upregulated by pro-inflammatory cytokines and therefore may be intimately linked to astrocyte reactivity [194]. As astrocyte reactivity is strongly integrated with NF-kB and YY1 upregulation, this would indicate that the suppressed capacity of these transcription factors to upregulate the astrocyte mitochondrial melatonergic pathway, will contribute to mitochondrial dysfunction, nuclear factor erythroid 2-related factor 2 (Nrf2)/GSH suppression, neuronal dysregulation, prolonged astrocyte reactivity and MHC-I linked autoimmune processes in both neurons and astrocytes, arising from suppressed paracrine and autocrine astrocyte melatonin. This may be important, as different cytokines act on astrocytes to determine their morphology and function, including the phenotypes of different immune cells chemoattracted into the CNS [195]. As depression is associated with a host of neurodegenerative and autoimmune conditions, processes acting to regulate the glia mitochondrial melatonergic pathway and its consequences for glia-neuronal interactions, form the basis of MDD comorbidities. See Figure 3.

The emerging importance of astrocytes in determining coordinated CNS function and the interface with wider systemic processes via the BBB, would indicate that core processes in astrocyte function are likely to be an integral aspect of understanding brain function. This article highlights how many of the diverse pathophysiological changes in many cells in MDD patients can involve alterations the regulation of the mitochondrial melatonergic pathway, including in CNS and systemic cells, with an array of diverse consequences. However, it is likely that astrocytes, as a core hub that integrates diverse CNS processes, coupled to their capacity to regulate neuronal activity and survival, as well as neurotransmitter release, oxidant and inflammatory challenges, will be an important treatment target for MDD and an array of wider medical conditions to which MDD is pathophysiologically associated.

This has a number of future research and treatment implications.

## 4. Future Research Directions

Does gut microbiome-derived butyrate require the capacity of the mitochondrial melatonergic pathway to be upregulated in order to mediate its extensive beneficial effects, especially in immune and glial cells?How does heightened circulating ceramide, linked to suppressed neurogenesis in MDD [96] and wider MDD pathophysiology [97], interact with mitochondrial melatonergic pathway in astrocytes? Is neurogenesis regulated by astrocyte NAS acting as a BDNF mimic at the receptor, TrkB?Does ceramide suppress 14-3-3ζ [196] and therefore the mitochondrial melatonergic pathway across different cell types, with consequences for the impact of gut microbiome-derived butyrate on mitochondrial and cellular function? As the over-expression of 14-3-3ζ prevents ceramide-induced autophagy [197], is this mediated via the upregulation of the melatonergic pathway?The importance of astrocytes, astrocytic mitochondrial melatonergic pathway, and the astrocyte network is paralleled in enteric glial cells. Enteric glial cells, like astrocytes, were classically conceptualized as providers of energy and antioxidants to enteric neurons. However, a growing body of data shows enteric glial cells to not only determine enteric neuron survival and function, but also to be an important interface with the gut microbiome, the mucosal immune system and vagal inputs to the gut. This may be important to MDD pathoetiology and its association with neurodegenerative conditions [198,199,200]. Do enteric glial cells have a mitochondrial melatonergic pathway, producing melatonin as shown in astrocytes, including regulation by apolipoprotein (Apo)E4 [170]? Are the effects of gut microbiome-derived butyrate in enteric glial cells dependent upon the capacity to upregulate the melatonergic pathway?Do NF-kB and YY1 induce the melatonergic pathway in astrocytes, leading to autocrine and paracrine effects that dampen inflammatory processes? Do endothelial cells, pericytes and systemic factors crossing the BBB regulate the astrocyte tryptophan-melatonergic pathway, thereby impacting on local neuronal regulation?Are the developmental regulatory effects of YY1 in astrocytes [175] dependent upon the capacity of YY1 to induce the melatonergic pathway?Are the detrimental effects of YY1 across a host of diverse medical conditions, including cancers, dementia, and type I diabetes, a consequence of melatonergic pathway suppression?Would the inhibition of the miRNAs that suppress 14-3-3 and the mitochondrial melatonergic pathway (e.g., miR-7, miR-375, miR-451 and miR-709) better optimize the beneficial effects of butyrate in MDD and related medical conditions?Is there an aging-associated increase in miRNAs that suppress the mitochondrial melatonergic pathway via the inhibition of the 14-3-3 isoforms, as with miR-709 in the murine liver [158]?Would the suppression of 14-3-3 isoforms and the mitochondrial melatonergic pathway have consequences for the regulation of the opioidergic system, e.g., the μ-/κ-opioid receptor ratio, and thereby impact on social processes as well as affective state? The integration of social processes, especially societal social stratification and discrimination, is challenging within a physiological framework. The above provides a basis for future investigations, including as to the relevance of a classical stress model, namely ‘stress-gut-dysbiosis-permeability-pro-inflammatory-cytokines--IDO-TDO-kynurenine-AhR’ pathway coupled to the differential opioidergic system regulation by butyrate and the mitochondrial melatonergic pathway. Would this be relevant to racial and sexual discrimination stress and its MDD and health disparities consequences?Some of the beneficial effects of melatonin are mediated via the alpha 7 nicotinic acetylcholine receptor (α7nAChR), including in the gut [201]. Are the gut barrier preserving effects of melatonin mediated via the α7nAChR on enteric glial cells?The α7nAChR, AhR and melatonin receptors are expressed on the mitochondrial membrane [87]. How do these receptors regulate mitochondrial function and how do they interact with the mitochondrial melatonergic pathway?Humans are unique in having duplicant α7nAChR (dupα7), which occludes α7nAChR effects. How relevant is this in cells linked to MDD pathophysiology, especially astrocytes, including in the regulation of cognition, given data showing its expression in distinct cortex regions in psychiatric conditions [202]?Future research investigating AhR effects in different cells, including astrocytes under different conditions and different AhR ligands, should better clarify the role of the AhR in MDD pathophysiology, including as to whether AhR effects vary according to the availability of the mitochondrial melatonergic pathway.Do early developmental impacts on the gut microbiome and gut permeability change the amygdala’s regulation of cortex and wider brain development, including inter-area interactions, with consequences for emotional and social processes? Does this involve alterations in the interactions of butyrate and the mitochondrial melatonergic pathway, including via the regulation of amygdala β-endorphin and μ-/κ-opioid receptor ratio, with consequent impacts on how the amygdala ‘affectively’ regulates other brain regions and their interaction.

Data from such future research will better clarify MDD pathophysiology and the role of the mitochondrial melatonergic pathway at the astrocyte-neuronal hub. It is not unlikely that astrocyte-neuronal interactions underpinning the subjective lowering of mood and alterations in cognition will be dependent upon the specific pathoetiology for a given individual. Triggers such as social stressors, opioidergic system dysregulation, gut dysbiosis/permeability, and early developmental stress will be driven by distinct physiological processes. However, it is likely that most, if not all, of these processes will have their impact suppressed by treatment targeted to the mitochondrial melatonergic pathway at the glia-neuronal interface at specific CNS sites. The above research is therefore likely to have significant treatment implications.

## 5. Treatment Implications

Given the powerful role of the gut in the pathoetiology and pathophysiology of MDD, the regulation of the gut microbiome and gut permeability has become a significant target for MDD treatment. As noted, enteric glial cells are significant determinants of enteric neuronal releases as well as providing an important interface for the gut microbiome and mucosal immune cells, platelets and vagal inputs. Recent work shows endocannabinoids to be significant inhibitors of enteric glial activation, with consequent benefits on the maintenance of the gut barrier. This would suggest the potential utilization of neutriceutical cannabinoids and cannabinoid-like molecules in the regulation of gut-linked MDD pathophysiology. The maintenance of the gut barrier will decrease levels of circulating LPS and therefore the activation of the LPS/NF-kB-YY1 pathway, which is problematic under conditions when neither NF-kB nor YY1 can induce the melatonergic pathway.Butyrate is one of the main beneficial factors produced by the gut microbiome, with butyrate producing bacteria being encouraged by pre- and pro-biotics as well as the dietary intake of foods that ferment in the large intestine, including complex carbohydrates. The nutriceutical, sodium butyrate, may also be useful as it provides an immediate increase in butyrate availability, whilst also encouraging the growth of butyrate producing bacteria [23]. As noted, such regulation of the gut microbiome/permeability has significant impacts on mitochondrial function across the body, with relevance to MDD pathophysiology.A number of studies have indicated the beneficial effects of α7nAChR agonists in MDD [203]. The α7nAChR is induced by melatonin and its utility in the gut, especially enteric glial cells and how they interface with vagal ACh inputs to the gut may be a relevant treatment target as well as the α7nAChR suppression of astrocytes and microglia reactivity [204].Alterations in platelet function may be relevant to both diagnosis and treatment in MDD, as previously proposed [205,206], including via alterations in serotonin uptake and release as well as in the regulation of ceramide, enteric glia and gut permeability.Clearly, early developmental processes are an important aspect of MDD pathoetiology. The above provides a research framework which should provide early development biomarkers and preventative interventions.

## 6. Conclusions

There is clear heterogeneity to MDD pathophysiology, as currently defined by non-physiological measures, which has considerably complicated an understanding of its pathoetiology and pathophysiology. This article has highlighted some of the processes relevant over the course of development that increase the likelihood of mood dysregulation, whilst emphasizing the importance of core physiological processes, namely the mitochondrial melatonergic pathway. The mitochondrial melatonergic pathway may be of particular importance in astrocytes, given their powerful regulation of neuronal activity, survival and neurotransmitter release. It is proposed that changes in the regulation of the glia mitochondrial melatonergic pathway underpin mood dysregulation and intercellular dyshomeostasis, whilst changing the impact that the gut microbiome-derived products can have on these processes. This leads to a number of novel research directions for MDD research from which better classification and treatment should emerge.

## Figures and Tables

**Figure 1 ijms-24-00350-f001:**
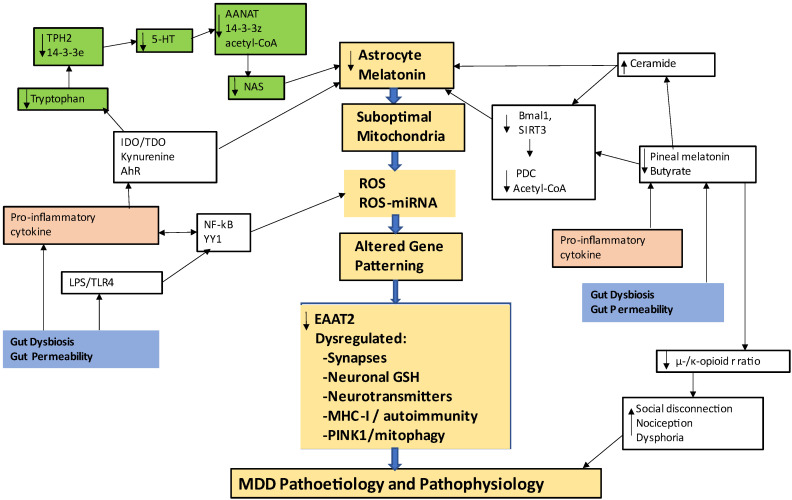
Shows how an array of different processes can impact mitochondrial function, highlighting factors regulating the astrocyte mitochondrial melatonergic pathway and its consequences for MDD pathophysiology (gold shade). The tryptophan-melatonin pathway is shaded in green. Tryptophan is taken up into cells where it is converted to serotonin by tryptophan hydroxylase (TPH2). In astrocytes, TPH2 requires 14-3-3ε for stabilization. Serotonin (5-HT) is converted to N-acetylserotonin (NAS) by AANAT, which requires 14-3-3ζ for stabilization, as well as acetyl-CoA. Factors suppressing the availability of these 14-3-3 isoforms will limit melatonergic pathway induction. The AhR and ceramide can suppress 14-3-3, thereby suppressing the melatonergic pathway. The AhR, purinergic P2Y1 and metabotropic glutamate receptor (mGluR)5 can ‘backward’ convert melatonin to NAS via O-demethylation (P2Y1 and mGluR5 not shown for clarity). Gut dysbiosis and gut permeability (blue shade) increase LPS and suppress butyrate, in conjunction with increased pro-inflammatory cytokines. Pro-inflammatory cytokines and stress/cortisol increase IDO and TDO, depleting tryptophan by converting it to kynurenine, which activates the AhR as well as leading to neuroregulatory kynurenine pathway products. Pro-inflammatory cytokines and LPS also suppress pineal melatonin, contributing to the loss of the circadian ‘resetting’ of mitochondria via pineal melatonin induction of Bmal1 and, like butyrate, sirtuin-3. Bmal1 and sirtuin-3 disinhibit PDC increasing pyruvate conversion to acetyl-CoA, which is necessary to induce the melatonergic pathway. The decrease in butyrate attenuates its induction of the μ-opioid receptor, thereby impacting on immune regulation, social processes and nociception. LPS and pro-inflammatory cytokines increase inflammatory factors via the transcription factors, NF-kB and YY1, which are normally dampened by their sequential induction of melatonin via melatonin’s intracrine, autocrine and paracrine effects. The impact of the gut and circadian dysregulation thereby has consequences for mitochondrial metabolism and increased ROS, leading to alterations in patterned miRNAs and consequently in patterned gene expression. Such alterations in astrocytes change neuronal regulation and transmitter release, with differential effects in distinct brain regions genetically and epigenetically primed by developmental processes. Abbreviations: AANAT: aralkylamine N-acetyltransferase; AhR: aryl hydrocarbon receptor; ASMT: N-acetylserotonin O-methyltransferase; CYP: cytochrome P450; IDO: indoleamine 2,3-dioxygenase; LPS: lipopolysaccharide; MHC-I: major histocompatibility complex-class I; NAS: N-acetylserotonin; NF-kB: nuclear factor kappa-light-chain-enhancer of activated B cells; PDC: pyruvate dehydrogenase complex; PINK1: PTEN-induced kinase 1; ROS: reactive oxygen species; TDO: tryptophan 2,3-dioxygenase; TLR: Toll-like receptor; YYI: yin yang 1.

**Figure 2 ijms-24-00350-f002:**
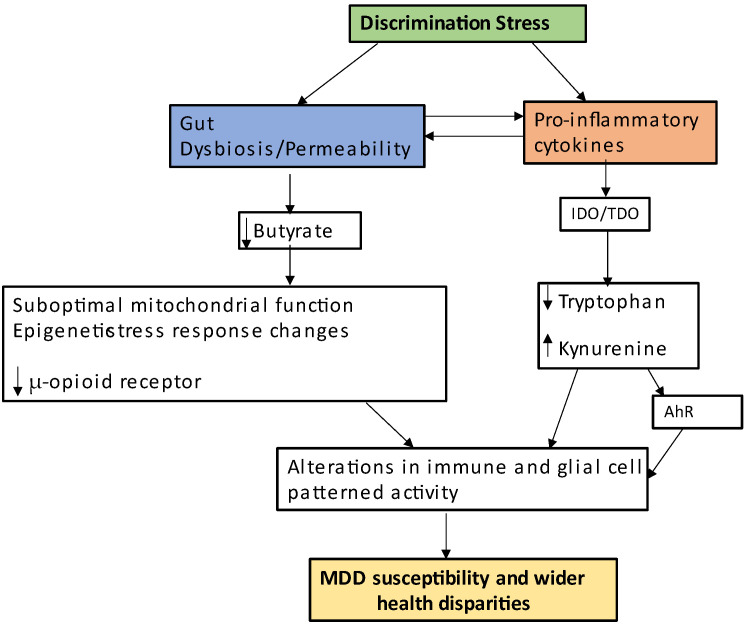
Discrimination stress increases gut dysbiosis and gut permeability as well as pro-inflammatory cytokines, which reciprocally induce each other. The pro-inflammatory cytokine induction of IDO/TDO decreases tryptophan/serotonin/melatonin levels, and consequently increases kynurenine. This has mitochondrial, AhR and patterned immune/glia activity consequences. Gut dysbiosis/permeability will also impact on mitochondrial function and patterned immune/glia responses, as well as on wider epigenetic regulation, coupled to a decrease in μ-opioid receptor. This has consequences for the health disparities evident across a range of medical conditions, including MDD. Abbreviations: AhR: aryl hydrocarbon receptor; IDO: indoleamine 2,3-dioxygenase; TDO: tryptophan 2,3-dioxygenase.

**Figure 3 ijms-24-00350-f003:**
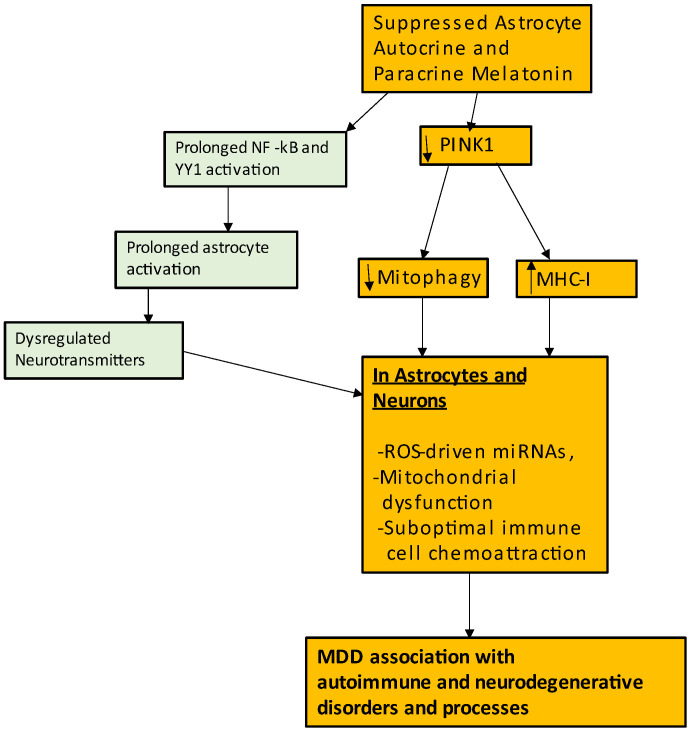
The suppression of astrocyte autocrine and paracrine melatonin also impacts on the regulation of processes associated with autoimmunity. Decreased autocrine and paracrine melatonin in astrocytes and neurons, respectively decreases mitochondrial membrane associated PINK1, leading to suppressed mitophagy of dysfunctional mitochondria and increased MHC-1. The associated increase in ROS alters patterned miRNAs and gene expressions, leaving cells more vulnerable to challenge. This may include alterations in the chemoattraction of immune cells. The astrocyte melatonergic pathway may therefore also be a major mediator of the association of MDD with autoimmune disorders as well as with autoimmune processes, such as MHC-1, which is highly expressed in the MDD brain. Abbreviations: MHC-I: major histocompatibility complex class I; NF-kB: nuclear factor kappa-light-chain-enhancer of activated B cells; PINK1: PTEN-induced kinase 1; ROS: reactive oxygen species; YY1: yin yang 1.

## Data Availability

Not applicable.

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
