# Peer review of "Depression Pathophysiology: Astrocyte Mitochondrial Melatonergic Pathway as Crucial Hub"

_ijms, 2022, doi:10.3390/ijms24010350_

Round 1
Reviewer 1 Report
The title of the article by G. Anderson would imply that astrocyte mitochondrial melatonergic pathways play a critical role in the etiology of major depressive disorders. The article itself seems only peripherally related to this topic. Astrocytes are really only introduced (at least more than shallowly) 14 pages into the article. This seems to be more an article about the role of the gut microbiome and inflammation. There is even a section on race, gender, and sexual identity descrimination.
Perhaps consider shortening the article and concentrating on one or two topics rather than making this an article about everything that has ever impacted melatonin,mitochondria, inflammation the gut, opiods, social processes, and actetylchonine (?).
Author Response
Manuscript ID: ijms-2046033
Type of manuscript: Review
Title: Depression Pathophysiology: Astrocyte mitochondrial melatonergic pathway as crucial hub
Reviewer 1
The title of the article by G. Anderson would imply that astrocyte mitochondrial melatonergic pathways play a critical role in the etiology of major depressive disorders. The article itself seems only peripherally related to this topic. Astrocytes are really only introduced (at least more than shallowly) 14 pages into the article. This seems to be more an article about the role of the gut microbiome and inflammation. There is even a section on race, gender, and sexual identity discrimination.
Perhaps consider shortening the article and concentrating on one or two topics rather than making this an article about everything that has ever impacted melatonin,mitochondria, inflammation the gut, opiods, social processes, and actetylchonine (?).
Response to reviewer 1
I agree. The manuscript has now been made more concise, with more of an emphasis on the topic of the themed edition, as suggested. However, it is important to embrace the complexity of factors that can impact on astrocyte regulation of neuronal activity.
The manuscript tries to embrace the complexity of data collected on MDD and challenge the limitations imposed by tight, but limited, paradigms for a condition that is generally accepted as poorly understood and treated. The manuscript has now been extensively rewritten to emphasize how the diverse bodies of collected data relevant to MDD may ultimately act on the glia-neuronal interface, of which the mitochondrial melatonergic pathway are an important aspect.

Reviewer 2 Report
Summary
The author provides a comprehensive review of how the mitochondrial melatonergic pathway in astrocytes might contribute to the pathophysiology of major depression. The author starts with a literature review of our current understanding of various aspects of astrocyte-related MDD pathophysiology, then attempts to connect these aspects into an integrated framework involving external stress, serotonin dysregulation, amygdala function, gut dysbiosis/permeability, mitochondrial melatonergic pathway, ROS, patterned gene expression, etc. Using this framework, the author proposes astrocytes as a physiological hub that coordinates MDD-related neuronal and functional changes. Some future research directions are listed to motivate filling the gaps in the proposed framework.
The quality of writing is good. The proposal being made seems pretty bold and might not turn out to be 100% the actual underlying mechanisms of MDD pathophysiology, but it contains many testable hypotheses that can motivate deeper research in this area to bring us closer to the biological underpinning of MDD. Thus, I believe this is a valuable piece of work.
Comments:
1. Mitochondria is ubiquitous in human cells and present in all neurons/astrocytes. I'd like the author to clarify why the astrocyte mitochondrial melatonergic pathway is especially important for MDD, but not other neuropsychiatric disorders.
2. It would be a good idea to estimate how much does mitochondrial melatonergic pathway contribute to MDD relative to the pure neuronal processes (i.e. if we have a hypothetical drug that prevents mitochondrial changes, how many percentages of MDD can we cure).
3. Figure 1 needs to be improved with better annotations. The up/down-regulation arrows look very similar to other arrows, which may lead to confusion. The fonts are also not very consistent.
4. In line 552, "This allows intercellular interactions to be conceived as a form on mitochondria to mitochondria communication". This seems to be an inaccurate description, oversimplifying the complexity of intercellular interactions, and overstating the significance of mitochondria communication. Same for line 562: "Cellular interactions may thereby be conceived as a form of modified bacteria communication with homeostasis being dependent upon all interacting cells being able to optimize their mitochondrial melatonergic pathway activity". In the end, mitochondria are just one of the many components in a human neuron/glia cell, and the physiological phenotypes depend on how the cells behave as a whole. I'd encourage the author to re-word these statements.
Author Response
Response to reviewers
Manuscript ID: ijms-2046033
Type of manuscript: Review
Title: Depression Pathophysiology: Astrocyte mitochondrial melatonergic pathway as crucial hub
Reviewer 2
The author provides a comprehensive review of how the mitochondrial melatonergic pathway in astrocytes might contribute to the pathophysiology of major depression. The author starts with a literature review of our current understanding of various aspects of astrocyte-related MDD pathophysiology, then attempts to connect these aspects into an integrated framework involving external stress, serotonin dysregulation, amygdala function, gut dysbiosis/permeability, mitochondrial melatonergic pathway, ROS, patterned gene expression, etc. Using this framework, the author proposes astrocytes as a physiological hub that coordinates MDD-related neuronal and functional changes. Some future research directions are listed to motivate filling the gaps in the proposed framework.
The quality of writing is good. The proposal being made seems pretty bold and might not turn out to be 100% the actual underlying mechanisms of MDD pathophysiology, but it contains many testable hypotheses that can motivate deeper research in this area to bring us closer to the biological underpinning of MDD. Thus, I believe this is a valuable piece of work.
Response to reviewer 2
Thank you for these encouraging comments.
Comments:
- Mitochondria is ubiquitous in human cells and present in all neurons/astrocytes. I'd like the author to clarify why the astrocyte mitochondrial melatonergic pathway is especially important for MDD, but not other neuropsychiatric disorders.
Response to reviewer 2
Thank you for mentioning this as it is an important point to clarify. The following text has been added to the manuscript: “The astrocyte-neuronal interface would seem to be an important hub in all neuropsychiatric disorders. However, the specificity of astrocyte-neuronal interactions in MDD would seem to be determined by particular interactions across brain regions, such as amygdala and cortex, over the course of development, and the influence of the gut microbiome on such interactions.”
- It would be a good idea to estimate how much does mitochondrial melatonergic pathway contribute to MDD relative to the pure neuronal processes (i.e. if we have a hypothetical drug that prevents mitochondrial changes, how many percentages of MDD can we cure).
Response to reviewer 2
This is an interesting point, although difficult to guestimate with confidence. The following has been added to the interface between Future Research and Treatment Sections:
“Data from such future research will better clarify MDD pathophysiology and the role of the mitochondrial melatonergic pathway at the astrocyte-neuronal hub. It is not unlikely that astrocyte-neuronal interactions underpinning the subjective lowering of mood and alterations in cognition will be dependent upon the specific pathoetiology for a given individual. Triggers such as social stressors, opioidergic system dysregulation, gut dysbiosis/permeability, and early developmental stress will be driven by distinct physiological processes. However, it is likely that most, if not all, of these processes will have their impact suppressed by treatment targeted to the mitochondrial melatonergic pathway at the glia-neuronal interface at specific CNS sites. The above research is therefore likely to have significant treatment implications. ”
- Figure 1 needs to be improved with better annotations. The up/down-regulation arrows look very similar to other arrows, which may lead to confusion. The fonts are also not very consistent.
Response to reviewer 2
Figure 1 has now been radically changed and is now clearer for the reader. Fonts have also been made consistent.
- In line 552, "This allows intercellular interactions to be conceived as a form on mitochondria to mitochondria communication". This seems to be an inaccurate description, oversimplifying the complexity of intercellular interactions, and overstating the significance of mitochondria communication. Same for line 562: "Cellular interactions may thereby be conceived as a form of modified bacteria communication with homeostasis being dependent upon all interacting cells being able to optimize their mitochondrial melatonergic pathway activity". In the end, mitochondria are just one of the many components in a human neuron/glia cell, and the physiological phenotypes depend on how the cells behave as a whole. I'd encourage the author to re-word these statements.
Response to reviewer 2
Thank you for highlighting this, as this is important to clarify. There is an overwhelming complexity of dynamic fluxes driving intercellular interactions. What I am trying to emphasize is the potential importance of core physiological processes driven by alterations in mitochondrial function. Any alteration in mitochondrial oxidant/antioxidant status will change ROS-driven microRNA, thereby impacting on patterned gene expression. Mitochondrial regulation, and the capacity to regulate the mitochondrial melatonergic pathway, is therefore a significant determinant of how cells interact and change each other’s gene expressions via dynamic intercellular fluxes. To date, most research has focussed on receptor activation, intracellular signalling pathways, and subsequent changes in the nucleus re gene expression. As a frame of reference re intercellular interactions, I suspect that this is where the ‘oversimplification’ has occurred. The capacity of cells to maintain optimized mitochondrial function, including a functioning melatonergic pathway, is an underestimated aspect of intercellular interactions that may be seen as akin to ‘resilience’ in MDD/stress paradigms. It would seem important to highlight the relevance of variations in mitochondrial function and redress this imbalance in core aspects of intercellular interactions. However, I take your point and have changed these two sentences to now read more tentatively, being an under-investigated aspect on intercellular interactions:
“This allows intercellular interactions to be at least partly conceived as a form on mitochondria-to-mitochondria communication”
“Cellular interactions may thereby be conceived, at least partly, as a form of modified bacteria communication with homeostasis being dependent upon all interacting cells being able to optimize their mitochondrial melatonergic pathway activity.”

Reviewer 3 Report
The paper is an admirable work. All etiopathogenetic mechanisms playing role in major depressive disorder, including the newest one (social stratification), are described in a clear and comprehensive way.
The summarization of treatment implications is rewarding especially for clinically oriented psychiatrists.
I see that the paper is result of a long-term effort and cooperation of interdisciplinary team.
I appreciate very much that the literature includes the newest publication. I have only one remark to the references. Mitochondria dysfunction especially in relation to depressive disorder is focus of interest of Fisar Z et al. Two papers have been published in the journal Mitochondrion with a high impact factor, and these should be included in the literature.
Author Response
Response to reviewers
Manuscript ID: ijms-2046033
Type of manuscript: Review
Title: Depression Pathophysiology: Astrocyte mitochondrial melatonergic pathway as crucial hub
Reviewer 3
The paper is an admirable work. All etiopathogenetic mechanisms playing role in major depressive disorder, including the newest one (social stratification), are described in a clear and comprehensive way.
The summarization of treatment implications is rewarding especially for clinically oriented psychiatrists.
I see that the paper is result of a long-term effort and cooperation of interdisciplinary team.
Response to reviewer 3
Thank you for these encouraging comments.
Reviewer 3: I appreciate very much that the literature includes the newest publication. I have only one remark to the references. Mitochondria dysfunction especially in relation to depressive disorder is focus of interest of Fisar Z et al. Two papers have been published in the journal Mitochondrion with a high impact factor, and these should be included in the literature.
Response to reviewer 3
Thank you for highlighting these two interesting references:
Fišar Z, Hansíková H, KÅ™ížová J, Jirák R, Kitzlerová E, Zvěřová M, Hroudová J, Wenchich L, Zeman J, Raboch J. Activities of mitochondrial respiratory chain complexes in platelets of patients with Alzheimer's disease and depressive disorder. Mitochondrion. 2019 Sep;48:67-77. doi: 10.1016/j.mito.2019.07.013. PMID: 31377247.
Hroudová J, Fišar Z, Kitzlerová E, Zvěřová M, Raboch J. Mitochondrial respiration in blood platelets of depressive patients. Mitochondrion. 2013 Nov;13(6):795-800. doi: 10.1016/j.mito.2013.05.005. PMID: 23688905.
Both have now been included in the manuscript and reference list, with the following text added: “Alterations in platelet function may be relevant to both diagnosis and treatment in MDD, as previously proposed [205-6], including via alterations in serotonin uptake and release as well as in the regulation of ceramide, enteric glia and gut permeability.”

Reviewer 4 Report
This paper has was written by an experienced author in the field of biological and pharmacological basis of mental disorders. Here the author tries to cover a broad range of detailed updated information regarding MDD in an educational textbook-like style. Consequently, although individual parts are well written, the overall description is diffuse and not well focused on the primary discussion point: astrocyte mitochondrial melatonergic pathways in MDD.
Specific points
1) Lines 238-265. The issue of social process and discrimination is critical point of MDD, but not much relevant to section 1 cellular and molecular level discussion of pathophysiology, and can be omitted.
2) Lines 306-311. The LPS stimulation-related cellular activation in MDD/stress-related disorders has been well characterized in microglial cells rather than astrocytes and these phrases are misleading.
3) Lines 470-485 The most of this part is previously discussed and seems redundant. Accordingly it should be rearranged or displaced.
4) Section 1.10 is missing.
5) Figure 1 is not well organized. Receptors and their down stream signal pathways of neurons, astrocytes and pineal endocrine cells are mixed up. The major pathophysiological roles of mitochondria in astrocytes is not explained well.
Author Response
Response to reviewers
Manuscript ID: ijms-2046033
Type of manuscript: Review
Title: Depression Pathophysiology: Astrocyte mitochondrial melatonergic pathway as crucial hub
Reviewer 4
This paper has was written by an experienced author in the field of biological and pharmacological basis of mental disorders. Here the author tries to cover a broad range of detailed updated information regarding MDD in an educational textbook-like style. Consequently, although individual parts are well written, the overall description is diffuse and not well focused on the primary discussion point: astrocyte mitochondrial melatonergic pathways in MDD.
Response to reviewer 4
Thank you. The manuscript tries to embrace the complexity of data collected on MDD and challenge the limitations imposed by tight, but limited, paradigms for a condition that is generally accepted as poorly understood and treated. The manuscript has now been extensively rewritten to emphasize how the diverse bodies of collected data relevant to MDD may ultimately act on the glia-neuronal interface, of which the mitochondrial melatonergic pathway are an important aspect.
Specific points
1) Lines 238-265. The issue of social process and discrimination is critical point of MDD, but not much relevant to section 1 cellular and molecular level discussion of pathophysiology, and can be omitted.
Response to reviewer 4
I agree that this is a relevant aspect of MDD. However, it is important to conceptualize such social processes as driving their impact via biological processes. Although data on this is still limited, I would like to maintain this within the section covering investigated biological underpinnings of MDD, as this is now a ‘hot topic’ of cutting-edge research. However, this has been significantly shortened, with removed parts shifted to the ‘Future Research’ section.
2) Lines 306-311. The LPS stimulation-related cellular activation in MDD/stress-related disorders has been well characterized in microglial cells rather than astrocytes and these phrases are misleading.
Response to reviewer 4
I have added the following to clarify this: “with effects mostly characterized to date in microglia, rather than astrocytes”
3) Lines 470-485 The most of this part is previously discussed and seems redundant. Accordingly it should be rearranged or displaced.
Response to reviewer 4
Thank you for highlighting this repetition. This section has been significantly reduced and repetitions removed.
4) Section 1.10 is missing.
Response to reviewer 4
Thank you for spotting this. Section numbering has now been corrected.
5) Figure 1 is not well organized. Receptors and their downstream signal pathways of neurons, astrocytes and pineal endocrine cells are mixed up. The major pathophysiological roles of mitochondria in astrocytes is not explained well.
Response to reviewer 4
Figure 1 has now been radically changed to be clearer to the reader.

Round 2
Reviewer 4 Report
Additional Figures seems to summarize the each critical points well, and in general, the text part of the revised article has became much more comprehensive.